# Effect of Hard- and Soft-Density Insoles on the Postural Control of Adults over 65 Years of Age: A Cross over Clinical Trial

**DOI:** 10.3390/bioengineering11121276

**Published:** 2024-12-16

**Authors:** Vicenta Martínez-Córcoles, Ricardo Becerro-de-Bengoa-Vallejo, César Calvo-Lobo, Eduardo Pérez-Boal, Marta Elena Losa-Iglesias, David Rodríguez-Sanz, Israel Casado-Hernández, Eva María Martínez-Jiménez

**Affiliations:** 1Department of Behavioral Sciences and Health, Miguel Hernández University of Elche, 03550 San Juan de Alicante, Spain; vicenta.martinezc@umh.es; 2Department Enfermería, Facultad de Enfermería Fisioterapia y Podología, Universidad Complutense de Madrid, 28040 Madrid, Spain; ribebeva@ucm.es (R.B.-d.-B.-V.); cescalvo@ucm.es (C.C.-L.); davidrodriguezsanz@ucm.es (D.R.-S.); isracasa@ucm.es (I.C.-H.); evamam03@ucm.es (E.M.M.-J.); 3Grupo FEBIO, Universidad Complutense de Madrid, 28040 Madrid, Spain; 4Department Enfermería y Fisioterapia, Universidad de León, 24401 Ponferrada, Spain; 5Department Enfermería y Estomatología, Universidad Rey Juan Carlos, 28922 Alcorcón, Spain; marta.losa@urjc.es

**Keywords:** insoles, balance, older adults, barefoot, hard, soft

## Abstract

Background: there is a high risk of falls in older adults. One of the factors contributing to fall episodes is advancing age due to deterioration of the proprioceptive system. Certain clinical procedures improve balance and posture, such as the use of insoles. Objective: the objective of this study was to evaluate the impact of hard and soft insoles on static foot balance in healthy older adults compared to barefoot people. Methods: a clinical trial was conducted with a sample size of 150 healthy adults (69 male and 81 women) with a mean age of 69.03 ± 3.82 years. Postural control was evaluated in different conditions of barefoot balance with hard and soft insoles. Results: All tests were statistically significant (*p* < 0.001). The test with soft insoles presented higher stabilometric values and presented worse postural control compared to the barefoot and hard insoles in all eyes-open and eyes-closed conditions. Conclusions: Hard and soft insoles show no improvement in postural control compared to barefoot standing. The findings suggest that soft insoles may result in impaired balance during standing. The density of the materials in the insoles emerges as a significant factor influencing postural control.

## 1. Introduction

Many of the injuries sustained from falls occur due to postural instability in the elderly population, thus posing a major health concern. As an adult ages, they become more vulnerable to the risk of falling, especially after turning 60 years old [1]. The deterioration in the proprioceptive system, which is a direct result of, is a key contributing factor in the increased incidence of falls amongst elderly people [2]. There is an increased risk of dynamic falls in younger people, compared to older adults who are more likely to suffer from static falls while stationary [3,4]. However, there are certain clinical interventions which are known to improve proprioception and posture. Furthermore, there is evidence that supports the use of footwear as a control mechanism in improving postural balance. For example, Finlay y colaboradores [5] evaluated gait in 60 healthy adult patients and observed that foot function, and consequently postural stability, improved when using a prescription shoe compared to those wearing their own footwear. Additionally, they observed a general improvement in the parameters used to assess gait whilst wearing the prescription shoes, due to the greater contact area with the ground which led to a decrease in forefoot loading and an increase in movement speed when compared to standard footwear. The features were personalized, intended to improve each individual’s current footwear conditions, and included the use of crossover slippers; extra-deep and extra-wide shoes; extra-wide shoes of normal depth; and shoes of different sizes. The study’s authors concluded that their findings supported the positive benefits of a special program of care consisting of footwear evaluation and modification [6].

Insoles have also been identified as a mechanism for improving postural control. Yan et al. [7] assessed the dynamic stability of 24 elderly individuals at low risk for falls. The authors demonstrated a more pronounced improvement in postural stability across whole-foot pressures. Priplata et al. [8] concluded that insoles improved somatosensory function and may be useful in alleviating age-related deficits in balance control.

Soft insoles have also been shown to have an impact on the electromyographic activity of the lower extremities, in terms of intensity, onset of muscular activity, and fatigue during dynamic and functional activities [9,10,11,12,13,14]. Although changes in muscular activity as a consequence of various insole densities indicate their improved effect on the plantar surface and tactile sensitivity of the feet, previous research has not specifically evaluated this in a vulnerable population of older adults.

One study, that investigated the effects of age and footwear on postural control concluded that shoes with soft and thick soles could impair stability by hindering the individual’s judgments relating to joint position. It is known that elderly patients often choose to wear slippers because of their soft material and flexible structure, which can comfortably accommodate painful feet and any potential deformities [15,16].

Unfortunately, soft-soled shoes can threaten the stability of older people because greater muscular activity is required to maintain stability when attempting to cease movement [17].

The link between postural stability and falling is well documented, with deficits in postural stability serving as significant predictors of falls amongst older adults. Therefore, it is imperative to identify and implement practical clinical interventions to improve postural stability in the elderly population [18,19].

To our knowledge, there has been no previous research describing the effects of insole density on standing postural stability. Therefore, the objective of this study was to evaluate the impact of two different shoe insoles on stationary balance in healthy older adults compared to those who went barefoot. We chose to evaluate postural sway in barefoot individuals, as opposed to those wearing shoes, because footwear has been shown to affect sensory feedback, potentially acting as a sensory filter between feet and the surface upon which one is standing [20,21,22]. We hypothesized that both types of insoles would decrease postural sway when compared to going barefoot.

In addition, there currently exist reliable and low-cost systems that can perform static stabilometric analysis quickly and efficiently, such as inertial sensors. Indeed, studies have carried out these measurements in young adults and children, but not in the elderly population. Consequently, we have decided to analyze the absolute and relative reliability of the Gyko^®^ system in the adult population [23] To evaluate the impact of two shoe insoles on stationary standing balance in healthy older adults compared to when going barefoot. The objective of the present study was to evaluate the impact of two shoe insoles on stationary standing balance in healthy older adults compared to when going barefoot.

## 2. Materials and Methods

In order to analyze the effect of insoles of different hard and soft densities on postural control in older adults, a cross over clinical trial was carried out from October 2023 to December 2023.

### 2.1. Subjets

This study involved men and women over 65 years of age who attend the Miguel Hérnandez University of Elche in San Juan, Alicante, for annual podiatry check-ups, who participated in this study. The sample size was calculated by measuring the differences between three paired groups according to the Wilcoxon signed-rank test via the G*Power (version 3.1.9.7 GmbH) software, with a two-sided hypothesis; with statistical power of 0.80, *p* < 0.5, an effect size of 0.50, and an error probability of 0.05; at 95% confidence interval, at s β error level of 20%, and a desired power analysis of 31 participants.

The type of insole was assigned randomly, providing each patient with an envelope containing the type of insole that they would use each time.

A group of 31 individuals who met the sample criteria randomly selected, and their participation and consent were requested. The 31 participants decided to participate in the study. The sample consisted of 19 healthy women and 12 men in the age range of 65 to 70 years (19 individuals, 61%) and 70 to 80 years (12 individuals 39%), with 61% in the age group of 65 to 70 years and 39% in the age group of 70 to 80 years (age 69 years ± 3.82; weight 70.45 kg ± 10.17; height 166.58 inches ± 7.4).

The criteria for inclusion were as follows: (a) The insoles involved were standard insoles which are not used to correct deformities; the differences between them concern the density of the materials (higher density is harder and lower density is softer). Participants could have any existing foot condition or deformity. Each pre-existing foot condition or deformity would be noted and listed for the sole purpose of evaluating the relationship between postural control and said deformity. (b) Participants have reached or are over 65 years of age, (c) present with normal or corrected vision, and (d) are capable of walking independently without the use of an assistive device, since the use of technical aids masks the potential effects of the insole.

The exclusion criteria were as follows: (a) present with a current injury, or injury 6 months prior to the test, in the lower limb, (b) suffer from a musculoskeletal disorder, (c) present with uncorrected vision, (d) are pregnant, (e) experience neurological disorders, diabetes, or lower limb amputation/prosthetics, plantar ulcers, or cognitive impairment [23].

### 2.2. Instrumental

To evaluate balance, participants were instructed to maintain the protocol anthropometric position proposed by the International Society for the Advancement of Kinanthropometry (ISAK); participants positioned their head on the Frankfort plane, with their upper limbs in a relaxed position, palms facing forward, and thumbs separated from the rest of their fingers.

Participants were required to stand barefoot, with their feet externally rotated at 30 degrees and with a distance of 4 cm between both heels [24].

A Gyko inertial sensor system (dimensions: 50 × 70 × 20 mm; mass: 35 g; Microgate Srl, Bolzano, Italy) (Image 1, 2 and 3) was used to collect the balance data (i.e., ellipse area (EA): length and surface area in cm^2^) (Figure 1) [25]. The sensor contains a three-axis accelerometer, gyroscope, and magnetometer which records (full scale range: 8 g) at a sampling rate of 500 Hz. During the evaluation, signals from the accelerometer and gyroscope were transferred via Bluetooth to a computer (HP Pavilion DV6, 15.6-inch, i7–3610QM 3rd Gen, 2.3 GHz, 4 GB RAM) and stored using the proprietary software v.1.2.2.0 (Gyko Re-Power Software). The software automatically calculated the length and surface projection, the velocity projection, and the frequency of oscillations. The Gyko system offers high reliability in measuring postural control compared to other measurement systems [24]. Previous research has shown that this protocol shows moderate to strong evidence of validity and reliability [25].

### 2.3. Method

This study was conducted according to the guidelines laid down in the Declaration of Helsinki, and all procedures involving human subjects were approved by the San Carlos Clinical Hospital Ethics Committee (Ref. 23/604-E) “Effect of hard and soft insole densities on postural control in adults over 65 years of age”. The participants of the study gave their informed consent, having received comprehensive and detailed information about the interventions involved.

The protocol of this study was publicly registered in ClinicalTrial.gov under the number NCT06634537.

All subjects completed three testing sessions in a laboratory setting without external distraction. The same testing procedures were repeated during each session, with a time period of one week between sessions. All participants were asked for their information (age, sex, weight, height, and date of birth), and they were then asked to take off their shoes in order to take the measurements. The Gyko^®^ device from Microgate, Spain was then placed in a harness on the back of the participant, who was subsequently asked to remain in an anthropometric position.

During the first testing session, postural impacts were assessed whilst subjects were barefoot. In the second testing session, subjects wore a soft gel insole: Cushioning gel slim (SIDAS, 18, rue Léon Béridot, Voiron, France: “URL (accessed on 10 January 2024)” https://www.sidas.com/es/plantillas-para-el-uso-diario/156-cushioning-gel.html). During the third and final test, Winter+ insoles were used (SIDAS, 18, rue Léon Béridot, Voiron, France; “URL (accessed on 10 January 2024)” https://www.sidas.com/es/plantillas-esqui-snowboard/221-winter-plus.html).

Each session lasted around 30 s, and data were collected while standing, with participants in a bilateral stance. Each task was performed both with eyes open and eyes closed. To control for possible variations in the visual field, subjects were asked to focus on a target placed 2 m in front of them at eye level. If the person moved or lost balance, the data were discarded and the test was repeated until it was obtained correctly.

In each test, the data were managed as follows: the first 10 s of each test were discarded; the average of the remaining 20 s was taken for later analysis. Postural sway was assessed using a set of measures. The sway area (cm^2^) was calculated using the area of the ellipse generated by the software. Additionally, sway distance and sway velocity were assessed along the anterior–posterior and medial–lateral axes.

### 2.4. Statistical Analysis

We assessed all variables for normality with the Shapiro–Wilk test and considered a variable to have a normal distribution if the probability value for a test was greater than or equal to 0.05.

To describe the demographic characteristics of the participants, we computed the means, standard deviations, and 95% confidence intervals of age, height, weight, and BMI.

We measured the intra-trial reliability with the three records for each CFI for each subject. To evaluate reliability within trials in each rider, we computed intra-class correlation coefficients (ICCs). For interpreting ICC values, we considered values less than 0.40 as poor, values between 0.40 and 0.59 as fair, values between 0.60 and 0.74 as good, and values 0.75 or greater as excellent [25]. Portney and Watkins [26] proposed that reliability coefficients greater than 0.90 were sufficient for clinical measurement. We also calculated the mean scores and the standard error of measurement (SEM). We used Brand and Altman’s [27] formula for the SEM as follows: SEM = SD × sqrt (1 − ICC).

The minimal detectable change (MDC) can be used to assess the minimal magnitude of change required to be 95% confident that the observed change between the 2 tests reflects the true change and not measurement error [28]. The MDC was calculated as 1.96 × SEM × √2. Paired Student *t* tests were used for parametric data and paired samples, and Wilcoxon’s test was used for non-parametric data in order to contrast the findings along the completed follow-up.

The statistical significance was established at *p* < 0.005, with a 95% confidence interval. The magnitude of differences was rated using Cohen’s d effect sizes. Substantial differences between hard and soft insoles were assumed when *p* < 0.05 and Cohen’s d > 0.80.

All data analysis was performed using IBM statistical processing (SPSS Statistics v.23.0 (SPSS Inc., Chicago, IL, USA).

## 3. Results

### Characteristics of the Total Population

Table 1 shows the anthropometric and demographic characteristics of the thirty-one healthy adults who participated in the study.

Table 2 shows excellent ICC values for all stabilometry variables in the different conditions, low SEM and MDC. Table 3 shows comparisons between the barefoot condition and hard and soft insoles. All tests were statistically significant (*p* < 0.001). The test with soft insoles presented higher stabilometric values and presented worse postural control compared to the barefoot and hard insoles in all eyes open and eyes closed conditions.

To find out if there were significant differences between hard and soft insoles, the Student *t* test was used to analyze the effect size and power in both conditions. The results are shown in Table 4.

## 4. Discussion

Sensory sensitivity exerts a significant influence on the maintenance of postural control. Vision, in particular, plays a fundamental role in postural balance. This is reflected in an observed increase in the ellipse area during the tests conducted without vision, a trend that had previously been observed in young adults [27,29], emphasizing the importance of this sense in the maintenance of postural stability [30,31].

Many older individuals resort to various insoles as support for walking, often without considering the potential repercussions on postural control. Therefore, this study aimed to evaluate the impact of insoles, hard and soft, compared to going barefoot, both weekly and biweekly. Postural control was examined while stationary with eyes open and closed. The hypothesis was that both types of insoles would improve postural balance compared to when barefoot. However, the results of the present study indicated that going barefoot showed better stabilometric data with a reduction in ellipse area and better postural control. In addition, it was observed that hard insoles offered results that were more similar to the barefoot position. On the other hand, soft insoles generated greater instability, which could increase the risk of falls in the vulnerable, elderly population.

The elimination of visual sensory information intensifies the demand on the proprioceptive system to preserve postural stability. This phenomenon manifested itself consistently in all evaluated conditions, both with the use of insoles, hard and soft, and whilst barefoot. It is important to note that participants whose eyes were closed and were using soft insoles exhibited the most unstable results on both interior–posterior and medial–lateral axes.

Significant improvements were observed whilst barefoot compared to wearing insoles, although these differences were more modest in the case of the hard insole, especially in the eyes-open condition, and more pronounced in the no-vision condition. There was a significant increase in stabilometric stability when standing with the soft insole compared to barefoot standing, reflected in the stabilometric values of the ellipse area.

The stiffer insole demonstrated superior stabilometric values compared to the soft insole but failed to surpass the stability observed in the barefoot condition. These results suggest to professionals prescribing orthopedic treatments that the use of a hard insole can be effective in preventing and reducing the risk of falls. This finding supports previous research that has pointed out the potential dangers of soft-soled footwear in older people, due to the associated muscular demand [24,32].

Although the insoles in the present study did not incorporate specific corrections, such as rearfoot reinforcement or arch support, previous studies have suggested that harder insoles may provide some control over pronation movement compared to softer insoles which adapt more to foot posture [33].

In summary, the present study suggests that, in orthopedic insole treatments, higher-density materials may be more effective in mitigating the risk of falls in older individuals, supporting findings in previous studies [34,35,36]. However, it is important to note that some studies, such as Halton’s, found no significant differences in older individuals when evaluating different insoles with various textures [37]. The approach differed from previous studies in both the inclusion of soft insoles and the participation of the geriatric population [38], along with the use of insoles and orthoses to improve postural control [39,40]

The present study is limited, to a certain extent, in its interpretation of the data. Firstly, participants’ foot types were not assessed, which could have influenced the results. In addition, measurements of ankle flexion and range of motion, a crucial factor in maintaining postural control, were not taken. Although differences were observed between types of insoles and the barefoot condition, prospective longitudinal studies will be needed to determine which stability variables are truly useful in predicting the risk of falls.

Furthermore, it will be essential to investigate whether the use of insoles or the type of material density can be effective in preventing falls over time. It will also be crucial to analyze the type of footwear in which such insoles would be used, since the design and characteristics of different footwear can have a significant impact on the effectiveness of the insoles when it comes to improving postural stability. These considerations are essential to fully understand the role of orthopedic insoles in preventing falls in the population studied.

## 5. Conclusions

The use of insoles, whether hard or soft, does not improve stationary postural control for older people compared to going barefoot. These results indicate that soft insoles, in particular, may lead to decreased balance while standing. The density of the materials in the insoles significantly influences posture control.

## Figures and Tables

**Figure 1 bioengineering-11-01276-f001:**
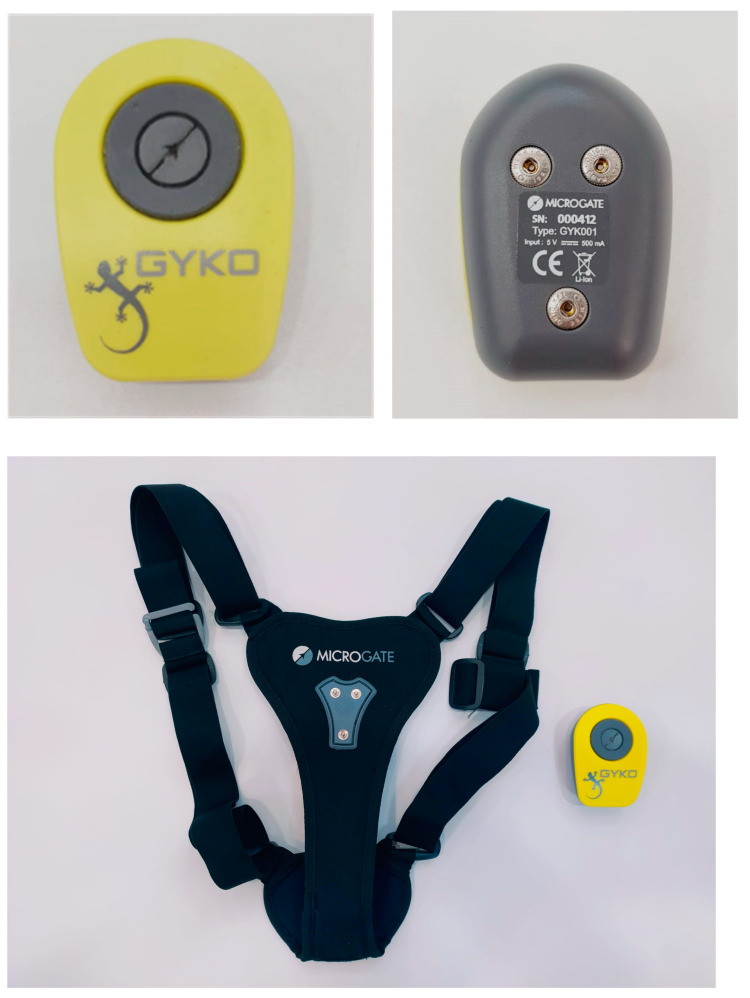
A Gyko inertial sensor system.

**Table 1 bioengineering-11-01276-t001:** Descriptive data of the participants total population.

Descriptive Data	Total Group	Male	Female	*p*-Value *
	Mean ± SD (95% CI) *n* = 31	Mean ± SD (95% CI) *n* = 12	Mean ± SD (95% CI) *n* = 19	
**Age**	**69.03 ± 3.82 (67.68–70.37)**	**70.83 ± 4.34 (68.37–73.29)**	**67.89 ± 3.05 (66.52–69.26)**	**<0.005**
Weight (kg)	70.45 ± 10.17 (66.87–74.03)	79.91 ± 7.78 (75.51–84.32)	64.47 ± 6.15 (61.70–67.24)	<0.001
Height (cm)	166.58 ± 7.40 (163.97–169.18)	172.75 ± 4.90 (169.97–175.52)	162.68 ± 5.94 (160.01–165.35)	<0.001
BMI (kg/m^2^)	25.27 ± 2.35 (24.44–26.10)	26.70 ± 2.21 (25.44–27.95)	24.36 ± 2.00 (23.46–25.27)	<0.005

Abbreviations: BMI, body mass index; kg, kilograms; SD, standard deviation; CI, confidence interval and * independent t student were applied. In all analyses, *p* < 0.05 (with a 95% confidence interval) was considered statistically significant.

**Table 2 bioengineering-11-01276-t002:** Intraclass correlation coefficients, standard error of measurement, and minimal detectable change and normality value for the ellipse area in stabilometry between barefoot, hard, and soft insoles.

Variable	ICC (95% CI)	SEM	MDC	*p*-Value
EA Open Eyes Barefoot	1.000 (0.999–1.000)	0.001	0.001	<0.001
EA Open Eyes Barefoot 7 d	1.000 (0.999–1.000)	0.001	0.001	<0.001
EA Open Eyes Barefoot 14 d	0.999 (0.998–0.999)	1.365	3.784	<0.001
EA Closed Eyes Barefoot	0.999 (0.998–0.999)	1.544	4.281	0.055
EA Closed Eyes Barefoot 7 d	0.999 (0.998–0.999)	1.555	4.312	0.144
EA Closed Eyes Barefoot 14 d	0.999 (0.998–0.999)	1.571	4.355	0.120
EA Open Eyes Hard insole	0.998 (0.996–0.999)	2.373	6.579	<0.025
EA Open Eyes Hard insole 7 d	0.999 (0.998–1.000)	1.675	4.643	0.038
EA Open Eyes Hard insole 14 d	0.999 (0.999–1.000)	1.68	4.657	0.049
EA Closed Eyes Hard insole	0.999 (0.999–1.000)	1.727	4.787	0.109
EA Closed Eyes Hard insole 7 d	0.999 (0.999–1.000)	1.779	4.933	0.099
EA Closed Eyes Hard insole 14 d	0.999 (0.998–0.999)	1.785	4.949	0.074
EA Open Eyes Soft insole	0.999 (0.998–0.999)	1958	5.429	0.200
EA Open Eyes Soft insole 7 d	0.997 (0.994–0.998)	3.404	9.437	0.200
EA Open Eyes Soft insole 14 d	0.999 (0.999–1.000)	1970	5.46	0.195
EA Closed Eyes Soft insole	0.997 (0.994–0.998)	3.802	10.53	0.200
EA Closed Eyes Soft insole 7 d	0.996 (0.992–0.998)	4.405	12.21	0.200
EA Closed Eyes Soft insole 14 d	0.989 (0.980–0.994)	7.438	20.61	0.174

Abbreviations: 95% CI, 95% confidence interval; ICC, intraclass correlation coefficient; SEM: standard mean average error; MDC: minimal detectable change; EA, ellipse area; 7 d; at seven days trial; 14 d, at fourteen days trial *p* value from Shapiro–Wilk’s normality test was applied. In all analyses, *p* < 0.05 (with a 95% confidence interval) was considered statistically significance.

**Table 3 bioengineering-11-01276-t003:** Stabilometric comparisons between the barefoot condition, hard, and soft insoles.

Variables	Barefoot		Hard Insole		Soft Insole		Barefoot vs. Hard Insole	Barefoot vs. Soft Insole
Variable	Mean ± SD (95% CI) Lim Inf-Lim Sup	Mean ± SD (95% CI)	Mean ± SD (95% CI) Lim Inf-Lim Sup	Median (IR)	Mean ± SD (95% CI) Lim Inf-Lim Sup	Median (IR)	*p* Value	*p* Value
EA Open Eyes	98.02 ± 42.85 (82.30 to 113.74)	97.37 (29.07)	123.25 ± 53.07 (103.78 to 142.72)	120.76 (54.14)	182.10 ± 61.94 (159.37 to 204.82)	186.79 (77.37)	<0.001	<0.001
EA Open Eyes 7 d	98.54 ± 42.80 (82.84 to 114.24)	96.10 (32.69)	123.38 ± 52.97 (103.95 to 142.82)	119.53 (56.02)	184.10 ± 62.16 (161.30 to 206.91)	185.62 (80.28)	<0.001	<0.001
EA Open Eyes 14 d	99.39 ± 43.18 (83.55 to 115.23)	96.38 (30.56)	123.27 ± 53.12 (103.78 to 142.75)	120.12 (52.79)	183.89 ± 62.30 (161.04 to 206.75)	186.04 (77.33)	<0.001	<0.001
EA Closed Eyes	116.86 ± 48.84 (98.95 to 134.78)	115.52 (47.03)	139.40 ± 54.62 (119.37 to 159.44)	136.92 (64.74)	208.53 ± 69.41 (182.53 to 233.46)	205.43 (122.50)	<0.001	<0.001
EA Closed Eyes 7 d	118.08 ± 49.19 (100.42 to 136.13)	115.25 (49.86)	140.61 ± 56.28 (119.96 to 161.25)	137.48 (60.94)	210.65 ± 69.66 (185.10 to 236.20)	206.97 (120.57)	<0.001	<0.001
EA Closed Eyes 14 d	118.29 ± 49.68 (100.06 to 136.51)	116.66 (48.19)	140.02 ± 56.46 (119.31 to 160.73)	137.01 (63.59)	208.22 ± 70.92 (182.20 to 234.23)	207.32 (121.15)	<0.001	<0.001

Abbreviations: SD: standard deviation; 95% CI: 95% confidence interval; IR: interquartile range; EA, ellipse area; 7 d; at seven days trial; 14 d, at fourteen days trial. Wilcoxon Signed Rank Test was applied. In all analyses, *p* < 0.05 (with a 95% confidence interval) was considered statistically significant.

**Table 4 bioengineering-11-01276-t004:** Differences between hard and soft insoles under different stabilometric conditions.

Variables	Hard Insole		Sotf Insole		*p* Value	1β	*d*
	Mean	SD	Mean	SD			
EA Open Eyes	**123.25**	**53.7**	182.10	31.9	<0.001	0.999	1.010
EA Open Eyes 7 d	123.38	53.97	184.10	62.16	<0.001	1	1.038
EA Open Eyes 14 d	123.27	53.12	183.89	62.30	<0.001	0.977	1.047
EA Closed Eyes	139.40	54.62	208.00	61.41	<0.001	0.999	1.083
EA Closed Eyes 7 d	140.61	56.28	210.65	69.66	<0.001	0.999	1.110
EA Closed Eyes 14 d	140.02	56.46	208.22	70.92	<0.001	0.980	1.63

Significant probabilities. Indicates the differences between hard and soft insoles *p* < 0.01 and *d* > 0.80.

## Data Availability

The datasets used and/or analyzed during the current study are available from the corresponding author on reasonable request.

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
