# Peer review of "Effect of Hard- and Soft-Density Insoles on the Postural Control of Adults over 65 Years of Age: A Cross over Clinical Trial"

_bioengineering, 2024, doi:10.3390/bioengineering11121276_

Round 1
Reviewer 1 Report
Comments and Suggestions for Authors
Thank you very much for the opportunity to analyze your work. I have the following comments:
- Please adjust the citation style so that quotes conform to the journal’s standards.
- In the context of sample size analysis, please revise the description to include the justification for the choice, power, and effect size.
- In the methods and materials section, please further justify the age range and heterogeneity of the group.
- For the analysis itself, please add an effect size analysis. First, in the description, include intervals for effect size, and in the results, provide effect size calculations alongside the p-values.
- The bibliography relies too heavily on outdated literature, with 25 sources being more than 10 years old, which is significantly too much. Please incorporate newer references.
I will assess the conclusions and discussion after the effect size has been added.
Author Response
Reviewer 1
Comment 1: Please adjust the citation style so that quotes conform to the journal’s standards.
Response 1: References to the American Chemical Society format have been modified.
Comment 2: In the context of sample size analysis, please revise the description to include the justification for the choice, power, and effect size.
Response 2: The following has been added to the statistical analysis section: L186 "The magnitude of differences was assessed using Cohen's d effect sizes. Substantial differences were assumed to exist between hard and soft insoles when p < 0.05 and Cohen's d > 0.80."
Comment 3: In the methods and materials section, please further justify the age range and heterogeneity of the group.
Response 3: The following has been added to the materials section “A group of 31 individuals were randomly selected, who were included in the sample criteria, where their participation and consent were requested. All 31 participants decided to participate in the study. The sample consisted of 19 healthy women and 12 men in the age range of 65 to 70 years (19 individuals, 61%) and 70 to 80 years (12 individuals 39%), with 61% for the age group 65 to 70 years and 39% for the age group 70 to 80 years (age 69 years ± 3.82; weight 70.45 kg ± 10.17; height 166.58 inches ± 7.4 ").
Comment 4: For the analysis itself, please add an effect size analysis. First, in the description, include intervals for effect size, and in the results, provide effect size calculations alongside the p-values.
Response 4: We have developed a table to better display the results of effect size and power. It is attached at the end of the document.
Comment 5: The bibliography relies too heavily on outdated literature, with 25 sources being more than 10 years old, which is significantly too much. Please incorporate newer references.
Response 5: We have updated 14 bibliographical references, it has been decided to keep some older references due to their importance and the little evidence of insoles in postural control.

Reviewer 2 Report
Comments and Suggestions for Authors
This is a well-written manuscript, minimal comments from me
L76: it looks like something is missing. The purpose of this study was to evaluate...
L91: The inclusion criteria...
Recommend re-doing Figure 2, they are blurry
Avoid the use of "we" or "our". Change from 1st person language to 3rd person language
Ethics - usually this section is part of the methods and comes usually in the opening paragraph
Results
avoid 1 sentence paragraphs
In tables change , to .
Author Response
Comment 1: L76: Something seems to be missing. The aim of this study was to assess...
Response 1: Added as reported by reviewer in L81: The aim of this study was
Comment 2: L91: The inclusion criteria...
Response 2: Described in L103
Comment 3: I recommend redoing Figure 2, they are blurry.
Response 3: We have added a new image to the paper.
Comment 4: Avoid using “we” or “our”. Change first person language to third person language
Response 4: Previously changed L210 y L214 “our” now “this”, L242 now “the result” L242“Our approach”, now L239 “The approach was different”, in L246 changed “our” to “the”, now in L243
Comment 5: Ethics: This section is usually part of the methods and usually appears in the opening paragraph.
Response 5: We have changed to L137 section 2.3 Method and study
Comment 6: Results: Avoid one-sentence paragraphs
Response 6: We have merged paragraph L196 into paragraph L197
Comment 7: In the tables it changes to,
Response 7: "to" has been removed

Round 2
Reviewer 1 Report
Comments and Suggestions for Authors
Thank you for the response. Most of the comments have been correctly addressed by the authors. However, the citation style does not meet the journal's requirements. Please correct it. Best regards.